# Animal Models of Chorioamnionitis: Considerations for Translational Medicine

**DOI:** 10.3390/biomedicines10040811

**Published:** 2022-03-30

**Authors:** Tiphaine Raia-Barjat, Margaux Digonnet, Antoine Giraud, Taghreed Ayash, Seline Vancolen, Mohamed Benharouga, Céline Chauleur, Nadia Alfaidy, Guillaume Sébire

**Affiliations:** 1Department of Gynaecology and Obstetrics, University Hospital of Saint-Etienne, 42055 Saint-Etienne, France; celine.chauleur@chu-st-etienne.fr; 2SAnté Ingénierie BIOlogie Saint-Étienne SAINBIOSE, DVH, INSERM U1059, Jean Monnet University, 42000 Saint-Etienne, France; antoine.giraud@chu-st-etienne.fr; 3Commissariat à l’Energie Atomique et aux Energies Alternatives (CEA), Biosciences and Biotechnology Institute of Grenoble, 38054 Grenoble, France; margaux.digonnet@cea.fr (M.D.); mohamed.benharouga@cea.fr (M.B.); nadia.alfaidy-benharouga@cea.fr (N.A.); 4Biosanté, MAB2 Team, Institut National de la Santé et de la Recherche Médicale U1292, 38000 Grenoble, France; 5Neonatal Intensive Care Unit, Department of Pediatrics, University Hospital of Saint-Etienne, 42055 Saint-Etienne, France; 6Child Health and Human Development Program, Child Health Department, Research Institute of McGill University Health Centre, Montreal, QC H4A 3J1, Canada; taghreed.ayash@mail.mcgill.ca (T.A.); selinevancolen@gmail.com (S.V.); guillaume.sebire@mcgill.ca (G.S.); 7Department of Obstetrics, CS 10217, University Hospital of Grenoble Alpes, 38043 Grenoble, France; 8Department of Pediatrics, McGill University, Montreal, QC H9X 3V9, Canada

**Keywords:** animal models, chorioamnionitis, preterm birth

## Abstract

Preterm birth is defined as any birth occurring before 37 completed weeks of gestation by the World Health Organization. Preterm birth is responsible for perinatal mortality and long-term neurological morbidity. Acute chorioamnionitis is observed in 70% of premature labor and is associated with a heavy burden of multiorgan morbidities in the offspring. Unfortunately, chorioamnionitis is still missing effective biomarkers and early placento- as well as feto-protective and curative treatments. This review summarizes recent advances in the understanding of the underlying mechanisms of chorioamnionitis and subsequent impacts on the pregnancy outcome, both during and beyond gestation. This review also describes relevant and current animal models of chorioamnionitis used to decipher associated mechanisms and develop much needed therapies. Improved knowledge of the pathophysiological mechanisms underpinning chorioamnionitis based on preclinical models is a mandatory step to identify early in utero diagnostic biomarkers and design novel anti-inflammatory interventions to improve both maternal and fetal outcomes.

## 1. Introduction

Chorioamnionitis (CA) has two definitions, namely acute (histological) and clinical CA, depending on whether the diagnosis is based on histological versus clinical criteria [1]. Acute CA is defined by histopathological criteria including chorion and amnion infiltration by neutrophilic polymorphonuclear leukocytes (PMNs) [2,3]. Clinical CA is a syndrome featured by more or less specific patterns of symptoms including maternal fever, uterine tenderness, malodorous leukorrhea, fetal tachycardia, maternal tachycardia, or maternal leukocytosis [2,4]. The sensibility of any combination of these elements to identify pathogen-induced intra-amniotic inflammation is limited to about 50% [5]. Mothers not fulfilling any criteria of clinical CA may carry fetuses affected by asymptomatic acute CA, which is diagnosed well after birth due to the delay inherent to placental delivery, handling, fixation and pathological exam. It has been proposed to replace the term “clinical CA” with a more general term, “intrauterine inflammation or infection or both”, abbreviated as “triple I” [6]. The term “suspected intrauterine infection” has also been suggested and defined as maternal intrapartum fever with one or more of the following: maternal leukocytosis, purulent cervical drainage, or fetal tachycardia [7]. Acute CA is more often due to bacterial infection in preterm than term delivery [2,8,9]. Acute CA is frequently polymicrobial due to a mix of aerobic and anaerobic bacteria originating mainly from the vaginal flora [10]. CA is associated with a sharp increase in the incidence of early neonatal bacterial infection and is a major cause of prematurity [11,12]. Importantly, exposure to CA is an independent risk factor for cerebral palsy (CP), and other non-CP neurobehavioral impairments [13]. Altogether, these observations indicate that CA is a threatening pregnancy pathology, which remains difficult to investigate in humans due to the lack of reliable global as well as causal prenatal diagnostic biomarkers. This is the reason why preclinical approaches remain currently the most valuable means to uncover the pathophysiological mechanisms of CA, early diagnostic markers, and to ultimately improve its treatment and outcome.

The choice of the relevant animal model to study CA and test potential therapies is essential as not all animal models recapitulate all features of this condition. Hence, the research question to be asked should be clearly targeted, while the animal model chosen must be appropriate, to optimize the translation of the findings to the human condition. In this line, the aim of this review is to compare the different existing animal models of CA with the goal of categorizing their use depending on the type of CA studied.

## 2. Epidemiology of Chorioamnionitis

Preterm birth (PTB) is commonly defined by the World Health Organization as the delivery of a viable fetus prior to 37 weeks of gestation. It affects 9.63% of live births [14]. Preterm birth is responsible for 70% of perinatal mortality and 50% of long-term neurobehavioral morbidities [11]. In women with premature labor, acute CA is found in 40% to 70% of cases based on ex vivo placental histology [15]. The prevalence of acute CA is approximately 5% of all deliveries, but this number masks large disparities depending on the gestational age. Indeed, the prevalence of acute CA is inversely proportional to the gestational age of delivery. CA affects nearly 40% of preterm deliveries between 25 and 28 weeks, and 4% of term deliveries [2]. Clinical CA is detected in 15% of cases in the antepartum and in 85% in the intrapartum period [16]. The presence of infectious agents induces an inflammatory response also by amniotic cells leading to the release of proinflammatory molecules such as prostaglandins into the amniotic fluid that could cause preterm labor occurring in the majority of chorioamnionitis pregnancies [17,18]. Importantly, preterm delivery is further favored by the disruption of cell junctions of fetal membranes making them prone to breakage [19].

The clinical situations that are associated with a high prevalence of acute CA are clinical CA (61%), spontaneous labor with intact membranes at term (6.3–18.8%) or preterm (8.7–34%), prelabor spontaneous rupture of membranes at term (34.3%) or preterm (17–57.7%), and prelabor premature rupture of membranes (PROM) with labor (75%) [2]. Other risk factors associated with acute CA are prolonged labor [20], prolonged membrane rupture [21], multiple vaginal exams [20], nulliparity [2,20], internal monitoring of labor [21,22], colonization with group B *streptococcus* (GBS) [20], and meconium-stained amniotic fluid [20,23]. Some other risk factors have been reported in smaller studies, such as smoking, alcohol or drug abuse, immune-compromised states, African-American ethnicity, epidural anesthesia, bacterial vaginosis, sexually transmissible genital infections, and vaginal colonization with ureaplasma [24].

CA is associated with a 2 to 3.5-fold increased risk of neonatal adverse outcomes, <34 and ≥34 weeks, respectively [25]. Perinatal death, neonatal sepsis, pneumonia, meningitis, cerebral hemorrhage, cerebral white matter damage, retinopathy of prematurity, necrotizing enterocolitis (NEC), bronchopulmonary dysplasia, and long-term disability including CP have been reported to be increased in the context of CA [26]. In premature infants, recent studies have demonstrated an association between exposure to CA and neurodevelopmental impairments from 18 to 30 months of corrected age [27,28,29], decreased cognitive performance at 5 years [30], autism spectrum disorder [31] and others. In the mother, CA has been shown to be associated with an increased risk for cesarean section deliveries, postpartum hemorrhage, endometritis, perineal infection, peritonitis, sepsis, and death [25,32].

## 3. Physiopathology of Chorioamnionitis

In clinical CA at both term and preterm, bacteria were identified in amniotic fluid in 61% [5] and 34% of women, respectively [5,33]. Preterm birth is associated with more frequent placental detection of pathogens, than term birth, with often polypathogen infections [34]. At term, the most frequent contaminating microorganisms are, in order of frequency, *Ureaplasma urealyticum*, *Gardnerella vaginalis*, *Mycoplasma hominis*, *GBS*, *Lactobacillus species*, and *Bacteroides species*. More than half of amniotic fluid cultures are positive for two or more bacteria [5]. Less likely, *Streptococcus anginosus*, *Escherichia coli*, *Candida species*, *Klebsiella pneumoniae* and *Listeria monocytogenes* have been detected [33]. In premature birth, *Ureaplasma species* are the most frequent microorganisms infecting the placenta. While these organisms are commonly part of the vaginal flora, aerobic vaginitis like *GBS* and *Escherichia coli* (*E. coli*) may induce an important host response and have been associated with ascending CA, PROM, and preterm birth [3]. The most frequent scenario of placental infection is ascending microbial invasion from the lower genital tract (Figure 1).

Other routes of contamination remain rare and include hematogenous, invasive procedures, or retrograde pathways via the fallopian tubes [11]. Vaginal organisms appear to ascend, first, in the cervix. Then, the adherent and virulent organism probably benefit from maternal immune tolerance during pregnancy to ascend to the lower pole of the uterus between the membranes and the chorion. The microorganism provokes a maternal inflammatory response at this level. The propagation continues through the placenta to the umbilical cord and up to the growing fetus through the fetal membranes up to the amniotic cavity. At this point, a fetal inflammatory response is initiated [2,35]. The Amsterdam Placental Workshop Group Consensus Statement proposed a staging and grading of the maternal and fetal inflammatory responses in ascending intrauterine infection, which is described in Figure 1 [1].

The control of the mechanism of parturition is very complex and involves, among others, the inflammatory system, namely inflammasome and/or toll-like receptors (TLR) pathways. The priming of the inflammasome leads to the activation of caspase-1 that triggers the final step of activation and release of interleukin (IL)-1β by the chorioamniotic membranes [36]. Beyond its participation in the acute phase response, IL-6 can also control inflammation by minimizing the impact of other inflammatory cytokines such as IL-1β and TNF-α [37]. Clinical investigations studying such biomarkers are feasible options, but provide only descriptive results with inherent limitations [38,39]. On the other hand, access to the fetus is difficult during gestation due to invasiveness. Hence, new therapies will emerge mostly from preclinical investigations.

## 4. Why Use Animal Models?

Numerous pathological processes that occur in pregnancy can be studied using freshly isolated trophoblast cells, trophoblastic cell lines, or placental explants [40,41,42]. However, such human material does not comprehensively reproduce the complex in vitro mechanisms. CA involves both maternal and fetal responses and is controlled by sophisticated cascades of immuno-inflammatory mechanisms. Such complex processes can be ideally uncovered at the mechanistic level in integrated systems, such as an animal model of CA. Also, the use of such preclinical models of CA allows for the testing of potentially placento-protective drugs to be subsequently used in phase I and phase II clinical trials in humans.

Most patients with clinical CA are diagnosed and treated with antibiotics well after the beginning of the disease—at advanced stages of infection—due to its initial, and often long lasting, subclinical phase [15]. This emphasizes the difficulty of addressing the study of this pathology at the subclinical stages in humans.

Hence, the use of animal models is mandatory to study the early stages of the disease. Animal models of CA have numerous advantages as they allow for: (1) identifying early biomarkers of CA, (2) testing antibiotic and anti-inflammatory drugs to prevent the noxious consequences of CA, such as preterm labor, (3) studying the pathogen-specific inflammatory responses at the feto–maternal interface, (4) identifying specific sets of inflammatory and microbiological markers at play for each pathogen, (5) determining fetal and neonatal short and long-term effects of CA, and their underlying mechanisms specific to each pathogen, and (6) testing novel therapeutic interventions to reduce inflammation and preterm birth, and consequently preserve perinatal and long term outcomes.

## 5. Animal Models of Chorioamnionitis

To date, numerous animals models of CA have been described in different species including rabbit [38,43,44,45,46,47,48], sheep [49,50,51,52,53,54,55,56,57,58], monkey [59,60,61,62,63,64,65,66,67,68,69,70], and guinea pig [71,72]. Nevertheless, murine models are the most commonly used in first line, mainly for practical and financial reasons. Hence, in this review, we have chosen to focus on the mouse and rat models of CA and focus on ascending route, rather than on invasion of the placenta by the hematogenous route, as seen in TORCH (Toxoplasmosis, Other Agents, Rubella, Cytomegalovirus, and Herpes Simplex) and others.

### 5.1. Differential Pregnancy Features in Actual Animal Models of Chorioamnionitis

It is well established that each species exhibits particularities regarding gestation. Term parturition in women occurs some 280 days after the onset of their last menstrual period. The mean time from ovulation to birth is 267 days (38 weeks, one day, standard deviation (SD): 10 days) [73]. In lower mammalian species (mouse, rat, rabbit), gestation is shorter—lasting between 20 and 22 days for rodents and 32 days for lagomorphs [74]. In sheep, the normal term is at approximately 150 days of gestation, 165 days for rhesus macaque, and 115 days for pigs [74]. Rodents, lagomorphs, and pigs have large litters; while humans, rhesus macaques and sheep often undergo singleton births [74].

In lower mammalian species and in sheep, term parturition occurs after involution of the corpus luteum and a subsequent decrease in serum progesterone. A systemic withdrawal of progesterone precedes labor in most species [35]. In contrast, a systemic progesterone withdrawal does not seem to be necessary for parturition to occur in non-human primates and in humans [35]. Similar to humans—rhesus macaques, rabbits, and rodents have discoid and hemochorial placenta. In sheep, the placenta is cotyledonary and epitheliochorial. In pigs, the placenta is diffuse and epitheliochorial [75]. Except for the macaques, all large laboratory animals exhibit important pregnancy discrepancies as compared to humans, including the type of the placenta. Rodents are less expensive than bigger animals, and allow for the use of many animals. Rodents tolerate surgeries, can be genetically modified and exhibit resistance to inflammatory stimuli, especially rats. Rats do not present preterm labor under inflammatory stress, such as lipopolysaccharide (LPS) from *E. coli* injection, the opposite to mice, which exhibit a high rate of preterm birth under the same conditions. Non-human primates have the most similar reproductive biology to humans and represent a near-ideal species in which to study CA and preterm birth. However, their high cost limits the number of animals to be included in given experiment [35]. Altogether, rodent models appear to be the optimal species to study CA, despite different reproductive biology.

### 5.2. Different Routes of Administration to Design Animal Models of Chorioamnionitis

The most frequent route that causes CA development in humans is the ascending microbial invasion from the lower genital tract. Hence, some murine models were developed through vaginal administration of bacteria (Table A1 and Table A2 in Appendix A). In mice with vaginal inoculation [76,77,78], the preterm birth rate varied from 27% to 54%. In a model with intracervical inoculation with endoscopy, the rate was 92% to 100% [79]. The low rate of preterm birth using vaginal inoculation can be explained by the superficial layers of the murine vaginal epithelium that are highly keratinized, which prevents bacterial adherence [78]. The two main studied routes for murine models are uterine horn injections upon mini-laparotomy or intraperitoneal injections [80]. The PTB rates are high in mice as they reach almost 100% for both routes. As for any administration, each route has its drawbacks. Intrauterine injection upon mini-laparotomy is conducted under anesthesia via surgery associated with significant maternal morbidity and mortality, and to intralitter heterogeneity in the level of exposure to pathogen components—or alive pathogens—and resulting inflammation, as compared to intraperitoneal injection [35]. Other routes have only been used punctually; these include ultrasound-guided intrauterine [81] or intra-amniotic [82,83] injection, intravenous [84,85,86,87] and intra-amniotic injection by laparotomy [88]. Intraperitoneal injection seems to be the optimal route of administration, according to animal ethics and consistency of the placental infection/inflammation.

### 5.3. Agents Used to Cause CA

CA can be induced by injecting different pathogen-associated molecular pattern molecules (PAMPs), damage-associated molecular patterns (DAMPs), or live microorganisms [89]. PAMPs are derived from microorganisms and thus drive inflammation in response to infections. The most common PAMP used to trigger CA is LPS [90,91] a component of the cell wall of Gram-negative like *E. coli*, *Salmonella enterica*, or *Salmonella typhimurium*. LPS does not appear to cross the healthy placental barrier [92] but this may change in inflammatory conditions. LPS activates the innate immune response by binding primarily Toll-like receptor 4 (TLR4) [93]. Others PAMPs used are those from killed and live *E. coli*, group B *Streptococcus* (*GBS*)—inducing mostly TLR2- and inflammasome-driven inflammation, and *Porphyromonas gingivalis*. Systemic exposure to LPS during the third trimester of gestation induced severe CA in dams, rapidly complicated by a decreased placental blood flow and placental infarcts showing that inflammation and thrombotic vasculopathy are tightly linked [94]. Interestingly, the LPS-induced CA is easily and early detected on in utero non-invasive magnetic resonance imaging, and significantly alleviated in terms of placental inflammation and cell death by the use of IL-1 blockade such as IL-1 receptor antagonist (IL-1ra) [89]. *GBS*-induced CA was particularly studied in a Lewis rat model using both inactivated *GBS* serotype Ia [95] or III [96] and live *GBS* serotype Ia [96,97]. Inflammatory responses in the placenta and the brain tissue as well as neurodevelopmental features of in utero-exposed progeny [95], sex-specific response [97], and effect of ampicillin treatment [98] were assessed in this model. A TLR2 agonist mimicking *GBS* infection was also used by other groups [91,99,100]. Inactivated versus alive *GBS* led to close placental and neurodevelopmental outcomes in the offspring, showing that the inflammatory response plays a key role in the physiopathological process leading to CA and its consequences in the progeny. Polyriboinosinic–polyribocytidilic acid (poly(I:C)), a viral double-stranded RNA mimetic, typically found in some viruses and activates Toll-like receptor 3 (TLR3), was also used experimentally to model viral infections in vivo [101,102]. One study assessed the administration of two PAMPs simultaneously in the same model (peptidoglycan (PGN) and poly(I:C) [103,104]. The effect of viral co-infection and bacterial ascension have been evaluated. These include *Ureaplasma urealyticum* before *E. coli* [76], *human influenza virus* [105], *murid herpesvirus 4* (MHV-68) [106]. Studies used synthetic lipopeptides (Toll-like receptor ligands) [107], or recombinant IL-1β [90,108] to further confirm the key roles of these specific molecules in the placental inflammatory response leading to CA.

In humans, more than half of specimen cultures collected from chorioamniotic tissues are positive for two or more bacteria with a wide variety of infectious agents. However, the organisms that are the most often isolated are not Gram-negative bacteria but ureaplasma and mycoplasm [80]. They are both wall-less bacteria known to activate the innate immune response through TLR2, 6, and 9. Few preclinical studies have already modelled ureaplasma-induced CA [83]. Animal models of CA, therefore, have limits since the bacteria used to develop them are not those most often involved in human CA [35]. Surprisingly, most preterm birth pregnancies do not exhibit positive cultures, indicating cryptic infection, sterile inflammation, or default of bacterial growth due to the prepartum administration of antibiotics in most women starting preterm labor [109,110,111]. DAMPs are endogenous intracellular molecules that are often released as a result of non-programmed cell death to convey danger cues in the first few hours of an injury; they are also referred to as alarmins [111]. DAMPs, such as uric acid, high mobility group box 1 (HMGB1), cell-free fetal deoxyribonucleic acid (DNA) (cffDNA), S100 proteins, heat shock protein 70 (HSP70), and adenosine triphosphate (ATP) have been reported to have a direct impact on the placenta in preclinical models and in humans [110]. Expression of these alarmins was increased in maternal serum or gestational tissue of women at risk of preterm labor [111]. Few, but important, studies successfully developed CA models via the administration of DAMPs such as uric acid crystals [112], cffDNA [113], alarmin S100A12 [114], or other alarmins such as HMGB1 [82,115]. DAMPs are also at play in rat models of CA and fetal inflammatory response syndrome induced by local hypoxia-ischemia, e.g., by transient (60 min) uterine artery occlusion [88]. These CA models accurately recapitulate key pathophysiological processes observed in extremely preterm infants, including placental, fetal, and brain inflammation [88].

### 5.4. Relevant Time Points of Infection to Develop Pertinent Rodent Models of Human CA

CA is a major cause of preterm birth in humans. In mice models of CA, the timing of induction of the infectious/inflammatory stress during gestation varies between 9 and 18.5 days of gestation (G) to mimic preterm or term CA. The most frequent time point used during gestation is G15 in mice, which corresponds to about 28 weeks of gestation (WG) for humans. To mimic end-gestational CA, studies were mostly conducted in rats by infectious/inflammatory stressors administered between G18 and 21 (normal rat gestation varies between G22–23), which corresponds to a level of brain development (immature myelination, cortical plate maturation and synaptogenesis) equivalent to 26–28 weeks of gestation in humans [116,117,118]. In this model, infectious agents are often injected at G18–19 for two reasons: (1) to mimic preterm onset of human CA, and (2) to investigate adverse effects of CA on neurodevelopmental processes pertinent for preterm human newborns. In fact, at G18–22 critical steps of glial cell development occurs in the rat brain [119], especially the final stages of oligodendrocyte differentiation, which are exquisitely vulnerable in the immature brain of preterm human newborns. Some models proposed preterm and term administration using inactivated GBS exposure repeated between G19 and G22 [97].

### 5.5. Differential Immune Responses in CA Animal Models

It is well established that the timing of the developmental process of the immune system in rodents is distinct from that of humans. Both hematopoiesis and differentiation of immune cells start in utero at 5 weeks of gestation in humans, in contrast to G8 in mice, corresponding to 10 weeks of gestation in humans [89].

The placenta is an important source of inflammatory cytokines and chemokines that are released from trophoblastic cells [120], infiltrating macrophages, and PMNs [121]. These proteins participate in the fetal inflammatory response [121]. However, it is still unclear whether the damaging fetal neuroinflammatory cascade originates from the fetal, placental, maternal compartment, or a combination of these three sources [122].

In mice, infectious agent inoculation provokes preterm birth within 24 h of administration. The steps of inflammatory responses are difficult to explore, given the speed in the occurrence of labor. In rats, the absence of preterm birth allows to conduct cesarean section at 24, 48, 72 h, and up to 5 days after administration. This possibility permits step-by-step analyses of inflammatory responses.

### 5.6. Therapeutic Approaches in Animal Models of Chorioamnionitis

The first approach is to reduce bacterial vaginal invasion through the use of vaginal hyaluronan [77] or poly(amidoamine) (PAMAM) dendrimers acting like [123] antibacterial agents. The second approach is to modulate inflammatory responses using TLR4 antagonist [124], antibody (Ab)-based depletion with Anti–Gr-1, anti–Ly-6G, or the appropriate IgG control Ab [125], IL-1β antagonist [90,94], regulatory T cells [126], recombinant IL-10 [127] or IL-6 [77,107,128,129,130], *N*-acetylcysteine [131], and interferon γ [77]. This also allows for the study of their impacts on placenta, fetal brain and neurodevelopment. A novel candidate is an analog of the PreImplantation Factor (PIF). The PIF modulates immune responses while reducing oxidative stress and protein misfolding [132]. The third approach is to use antibiotics such as amoxicillin to directly act on bacteria. The fourth approach is the use of treatments avoiding preterm birth, such as progesterone. Finally, drugs such as magnesium sulfate are used for fetal and neonatal neuroprotection.

## 6. Translational Medicine Perspectives

Nowadays, clinicians are still facing diagnostic challenges when it comes to CA. The diagnosis of CA implies that a pregnant woman has an inflammatory or an infectious disorder of the chorion, amnion, or both [7]. To avoid this confusion, the latest recommendations propose to abandon the use of the term CA in favor of, Intrauterine Inflammation or Infection, or both, abbreviated as Triple I.

The criteria entering the new definition are mainly clinical. Triple I is categorized as suspected, without confirmation of infection or confirmed, often retrospectively, upon laboratory analyses that demonstrate infection in the amniotic fluid, or upon histopathological analyses showing infection or inflammation in the placenta, fetal membranes, or in the umbilical cord vessels (funisitis). Clinicians most often face three situations before the diagnosis of the disease: (1) a patient presenting with labor and fulfilling all the criteria of triple I. In this case, CA is declared, and the diagnosis does not pose any problem, (2) patient presenting either with a PROM with an expectation of CA risk or preterm labor or (3) patients with unrecognized subclinical CA.

Given the uncertainties about the early and accurate diagnosis of CA, there is a critical need for the discovery, validation, and implementation of preclinical and clinical studies. These would be beneficial to identify reliable biomarkers that could stratify women regarding the risk and subtype of CA and subsequent specific fetal and neonatal complications. These biomarkers would aid in the decision to transfer high-risk women to maternity hospitals offering level and profiles of cares adapted to their gestational age. They would also allow for optimal treatment with antibiotics or anti-inflammatory drugs such as steroids, or other novel compounds, personalized for each woman depending on the pathogen and/or sterile trigger(s). The diagnosis of CA implies the birth of the fetus to keep it away from inflammation or infection, or both. Hence, reliable biomarkers would allow for decision-making in response to the following questions: conservative management or delivery, use of tocolytic treatment, conductance of cervical cerclage in the case of high infectious risk for the fetus [6]. To date, markers associated with inflammation in the maternal blood have been investigated but have failed to show clinical utility [133,134]. Researchers have analyzed the amniotic fluid in order to study the gestational sac and not the maternal systemic [135]. The utility of this invasive procedure remains unsettled but was proposed in cases of asymptomatic short cervix [6]. Preclinical models bring hope to the use of non-invasive imaging (magnetic resonance) of the placenta for an early diagnosis of CA [94]. Novel methods are being developed to aid detection of inflammation like optoacoustic which is a technology for non-invasive visualization of laser-illuminated tissue by the detection of acoustic signals [136]. Unfortunately, such an approach still needs validation in human studies.

The standard treatment for clinical CA is the administration of antibiotics agents and delivery. Concerning antibiotic use, current recommendations are based on one randomized controlled trial comparing initiation of the treatment in the intrapartum period versus immediately after delivery [137]. A survey on practice reports the use of over 25 different primary antibiotic regimens in clinical practice [138]. Surprisingly, one preclinical study used amoxicillin in a CA animal model and found an increase in placental inflammation. The authors hypothesized a potential link between antibiotic-induced bacterial lysis and related inflammatory surge. Subsequently, they proposed to explore the combined effects of anti-inflammatory treatments and antibiotic therapy [98].

Among other reliable treatments that act on the fetal side are antenatal corticosteroids and magnesium sulfate, known to ensure fetal neuroprotection in the case of preterm birth. However, in clinical CA, recommendations are not unanimous worldwide regarding the use of corticosteroids, given the associated immunosuppressive side effects. Immediate delivery after the diagnosis of clinical CA does not definitely prevent adverse maternal and neonatal outcomes or long-term neurodevelopmental outcomes, but delivery if labor is not in course must be considered. The administration of at least one dose of corticosteroids has a beneficial effect on the neonate without increasing the risk of sepsis or other adverse neonatal outcomes [139]. The administration of antenatal magnesium sulfate is also beneficial in clinical CA with preterm birth, as it reduces the prevalence of CP [6]. In regards to placento- and neuro-protection, many studies using animal models of CA proposed to modulate the inflammatory cascade with the use of IL1 blockade [90,94,140,141]. This treatment was successfully administered before or after injection of the infectious agent [109,110,111]. This brings us to the threshold of preclinical data required for proposing human phase II randomized controlled trials using human recombinant IL-1Ra as a placento- and neuro-protective drug. This would be a repurpose of this FDA-approved drug, already used with excellent safety in chronic inflammatory diseases, including in pregnant women and newborns. The study will aim to confirm preclinical data showing that IL-1 blockade administered to the mother and/or newborn will alleviate the heavy burden of multisystemic developmental disabilities arising from bronchopulmonary dysplasia, NEC, and perinatal white matter injuries all caused by the IL-1-driven fetal inflammatory response arising from CA [140,142]. Neonatal management also evolves with a reduction in antibiotic use [143] and the exploration of new placento- and neuro-protective interventions.

## 7. Conclusions

Developing new biomarkers or therapies for CA ultimately requires additional preclinical studies that are based on reliable animal models and/or RCT testing very promising compounds such as IL-1 blockade. These models will allow for testing both the efficacy and safety of therapeutic candidates. The choice of the most relevant animal model will need to take into account the characteristics of pregnancy, particularly the animal species, the route of agent administration, and the type of agent to induce inflammation. It is important to frame the question to be asked and the model to be used, as this will influence the inflammatory pathway, at the maternal, fetal and neonatal levels and will therefore impact the translation and applicability of the findings to humans. Because CA involves the mother and the growing fetus, with a validated impact on the development of its brain, preclinical models that consider both the impact of the inflammation on the placental and fetal development must be prioritized for future research on CA.

## Figures and Tables

**Figure 1 biomedicines-10-00811-f001:**
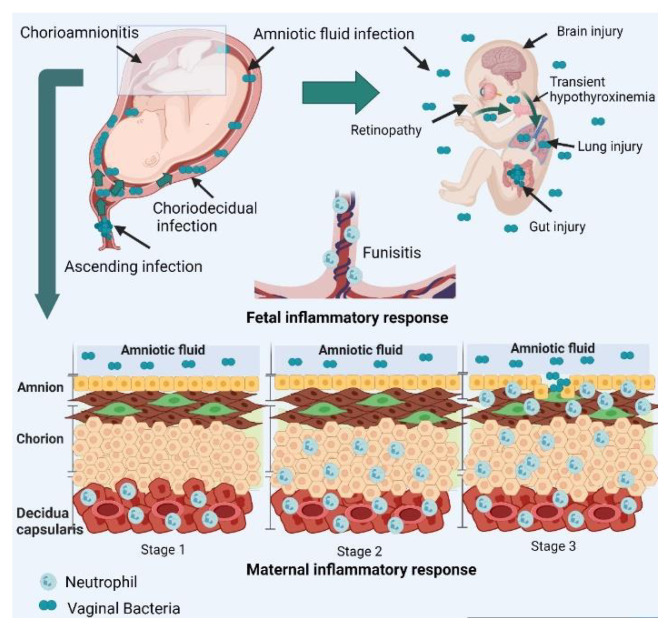
Physiopathology of CA at the maternal and fetal sides. Maternal inflammatory response is defined as stage 1 in case of acute subchorionitis or chorionitis, stage 2 in case of acute CA: polymorphonuclear leukocytes extend into fibrous chorion or, amnion and stage 3 in case of necrotizing CA: karyorrhexis of polymorphonuclear leukocytes, amniocyte necrosis, and/or amnion basement membrane hypereosinophilia. Fetal inflammatory response is defined as stage 1 in the case of chorionic vasculitis or umbilical phlebitis, stage 2 in case of involvement of the umbilical vein and one or more umbilical arteries, and stage 3 in case of necrotizing funisitis [1]. (Created with BioRender.com 23 March 2022).

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
