# Peer review of "Animal Models of Chorioamnionitis: Considerations for Translational Medicine"

_biomedicines, 2022, doi:10.3390/biomedicines10040811_

Round 1

Reviewer 1 Report

In this review authors summarized recent advances in the understanding of the underlying mechanisms of chorioamnionitis and subsequent impacts on the pregnancy outcome also describing relevant and current animal models of chorioamnionitis used to decipher associated mechanisms and develop much needed therapies. 

the manuscript is clear and well written. However, some points need to be improved:

  • 2. Epidemiology of chorioamnionitis: Authors must explain more deeply how CA can lead to preterm delivery. In particular, should be specified that the presence of infectious agents induces an inflammatory response also by amniotic cells leading to the release of proinflammatory molecules such as prostaglandins into the amniotic fluid that could cause preterm labour occurring in the majority of chorioamnionitis pregnancies (PMID: 20331587, 20031205). Importantly, preterm delivery is further favored by the disruption of cell junctions of foetal membranes making them prone to breakage (PMID: 26739007). 
  • Line 119:  What is PPROM? do the authors mean PROM?
  • Figure 1: the term "decidua" must be replaced with "decidua capsularis" otherwise it may be confused with the decidua basalis not involved in the foetal membrane since is the "maternal" part of placenta.  

Author Response

Thank you very much for your positive decision regarding our paper “Animal models of chorioamnionitis: Considerations for translational medicine”. We have carefully read the Reviewers comments and are pleased to submit a revised version of the manuscript in which most of their questions have been taken into consideration.

Response to reviewers

Reviewer 1

In this review authors summarized recent advances in the understanding of the underlying mechanisms of chorioamnionitis and subsequent impacts on the pregnancy outcome also describing relevant and current animal models of chorioamnionitis used to decipher associated mechanisms and develop much needed therapies. 

the manuscript is clear and well written. However, some points need to be improved:

  • 2. Epidemiology of chorioamnionitis: Authors must explain more deeply how CA can lead to preterm delivery. In particular, should be specified that the presence of infectious agents induces an inflammatory response also by amniotic cells leading to the release of proinflammatory molecules such as prostaglandins into the amniotic fluid that could cause preterm labour occurring in the majority of chorioamnionitis pregnancies (PMID: 20331587, 20031205). Importantly, preterm delivery is further favored by the disruption of cell junctions of foetal membranes making them prone to breakage (PMID: 26739007). 

We have added these precisions which indeed clarifies the text

  • Line 119:  What is PPROM? do the authors mean PROM?

We have made this correction.

  • Figure 1: the term "decidua" must be replaced with "decidua capsularis" otherwise it may be confused with the decidua basalis not involved in the foetal membrane since is the "maternal" part of placenta.  

We have made this correction.

Reviewer 2 Report

The authors present a comprehensive review

of animal models used in chorioamnionitis.

The level of detail presented for the different topics

offered a good readability and enabled a quick

overview. 

This reviewer only has minor issues to be addressed:

1) Similar to limiting your review to rodent models, it would

be helpful if the authors would state that

this review does focus on ascending, rather than

on invasion of the placenta by the hematogenous

route, as seen in TORCH…

2) Typo: Line 198: discord—>discoid?

3) Line 370: non-invasive imaging

(magnetic resonance)of the placenta

for an early diagnosis of CA—>

The authors could also state that novel methods

are being developed that aid detection of inflammation,

e.g.
Optoacoustic Imaging in Inflammation
Adrian P. Regensburger er al. J.

Biomedicines 2021, 9, 483.

https://doi.org/10.3390/biomedicines9050483

Author Response

Thank you very much for your positive decision regarding our paper “Animal models of chorioamnionitis: Considerations for translational medicine”. We have carefully read the Reviewers comments and are pleased to submit a revised version of the manuscript in which most of their questions have been taken into consideration.

Response to reviewers

Reviewer 2

The authors present a comprehensive review of animal models used in chorioamnionitis.

The level of detail presented for the different topics offered a good readability and enabled a quick

overview.  This reviewer only has minor issues to be addressed:

1) Similar to limiting your review to rodent models, it would be helpful if the authors would state that

this review does focus on ascending, rather than on invasion of the placenta by the hematogenous

route, as seen in TORCH…

We have added these precisions

2) Typo: Line 198: discord—>discoid?

We have made this correction.

3) Line 370: non-invasive imaging (magnetic resonance)of the placenta for an early diagnosis of CA—> The authors could also state that novel methods are being developed that aid detection of inflammation, e.g. Optoacoustic Imaging in Inflammation Adrian P. Regensburger er al. J. Biomedicines 2021, 9, 483. https://doi.org/10.3390/biomedicines905048

We have added these precision and the reference, thanks.
